# Solving Systems of Linear Equations: HHL from a Tensor Networks Perspective

**Alejandro Mata Ali**[1*], **Iñigo Perez Delgado**[2†], **Marina Ristol Roura**[2‡], **Aitor Moreno Fdez. de Leceta**[3∘] **and Sebastián V. Romero**[4§]

**1** i3B Ibermatica, Quantum Development Department, Paseo Mikeletegi 5, 20009 Donostia, Spain
**2** i3B Ibermatica, Parque Tecnológico de Bizkaia, Ibaizabal Bidea, Edif. 501-A, 48160 Derio, Spain
**3** i3B Ibermatica, Unidad de Inteligencia Artificial, Avenida de los Huetos, Edificio Azucarera, 01010 Vitoria, Spain
**4** TECNALIA, Basque Research and Technology Alliance (BRTA), 48160 Derio, Spain

⋆ alejandro.mata.ali@gmail.com , † iperezde@ayesa.com , ‡ mristol@ayesa.com ,
∘ aitormoreno@lksnext.com , § sebastian.vidal@tecnalia.com

## Abstract

This work presents a new approach for simulating the HHL linear systems of equations solver algorithm with tensor networks. First, a novel HHL in the qudits formalism, the generalization of qubits, is developed, and then its operations are transformed into an equivalent classical HHL, taking advantage of the non-unitary operations that they can apply. The main novelty of this proposal is to perform a classical simulation of the HHL as efficiently as possible to benchmark the algorithm steps according to its input parameters and the input matrix. The algorithm is applied to three classical simple simulation problems, comparing it with an exact inversion algorithm, and its performance is compared against an implementation of the original HHL simulated in the Qiskit framework, providing both codes. It is also applied to study the sensitivity of the HHL algorithm with respect to its hyperparameter values, reporting the existence of saturation points and maximal performance values. The results show that this approach can achieve a promising performance in computational efficiency to simulate the HHL process without quantum noise, providing a higher bound for its performance.

# 1  Introduction

The solution of linear equation systems $A\vec{x} = \vec{b}$ is a fundamental problem in many areas of science and engineering. Classical methods for solving these equations, such as *Gaussian elimination* and *LU decomposition* [1], have been widely used and optimized for decades [2,3]. However, as the size of the system grows, classical methods become computationally expensive and inefficient. One of the most efficient classical methods is the *conjugate gradient method* (CG) [4,5], which has a complexity of $\mathcal{O}\left(Ns\kappa \log\left(\frac{1}{\epsilon}\right)\right)$ for a matrix $N \times N$ with a maximum of $s$ non-zero elements per row, $\kappa \equiv \frac{|\lambda_{max}|}{|\lambda_{min}|}$, $\lambda$ being the eigenvalues of the matrix $A$ and $\epsilon$ the error.

Quantum computers offer the potential to solve some challenging problems more efficiently than classical computers. In particular, the *HHL algorithm* proposed by Harrow, Hassidim, and Lloyd in 2008 [6,7] is a method for solving linear equations that runs in polynomial time, where the polynomial depends logarithmically on the size of the system. It is intended for the calculation of the expectation values in $\mathcal{O}\left(\log(N)s^2\kappa^2/\epsilon^3\right)$, as it loses its advantage in the case of extracting the explicit solution, which is produced in time $\mathcal{O}\left(\log(N)s^2\kappa^2/\epsilon\right)$. This algorithm has three main hyperparameters: $\tau$ the evolution time, which rescales the eigenvalues of the problem, $n_c$ which determines the precision in the eigenvalues, and $C$ which rescales the eigenvalue inversion step. The dependence of the algorithm performance with respect to these three hyperparameters, even in the noise-free regime, is an important open question.

However, the current Noisy intermediate-scale quantum (NISQ) state of the quantum hardware limits the testing of this algorithm, restricting its benchmarking to classical simulators. For the HHL algorithm, the most feasible simulation framework is the statevector simulation. This framework has several limitations, such as the exponential increase in the computational resources required with respect to the number of qubits, which is approximately $\mathcal{O}\left(N^2\left(\frac{\kappa}{\epsilon}\right)\log\left(\frac{\kappa}{\epsilon}\right) + N\left(\frac{\kappa}{\epsilon}\right)^2\right)$ in the best possible case and $\mathcal{O}\left(N^2\left(\frac{\kappa}{\epsilon}\right)\log\left(\frac{\kappa}{\epsilon}\right) + N\left(\frac{\kappa}{\epsilon}\right)^3\right)$ in the most common situation (as proved in Sec. 2). This is due to the simulation of a $n + n_c$ qubit circuit with a general Hamiltonian evolution and the $n_c$ steps of the quantum phase estimation, $n = \lceil \log_2 N \rceil$ and $n_c = O(\log_2(\kappa/\epsilon))$ being the number of clock qubits required for the precision of the eigenvalues.

Recently, there has been growing interest in using qudits [8] and tensor networks [9,10] to implement different quantum algorithms. Qudits are generalized qubits with more than 2 basis

states. Tensor networks are classical representations of tensor algebra equations, providing an efficient way to represent and manipulate certain types of high-dimensional systems, such as quantum states with low entanglement [11], or compressing machine learning models [12, 13], enabling quick computations with classical computers [14–16]. The tensor networks framework is also applied to circuit simulation [17], applying tensor compression, but it scales exponentially with respect to the entanglement. Moreover, if it is applied directly without other optimizations adapted specifically to the circuit simulated, the amount of resources can scale even more, making it even slower than the statevector simulation.

In this paper, a novel approach is proposed for simulating the HHL linear systems of equations solver algorithm using qudits and tensor networks, providing an efficient way to test the higher bound in the possible performance of the algorithm. This algorithm runs in $\mathcal{O}(N\kappa^2/\epsilon^2 + N^2\kappa/\epsilon + \kappa^3/\epsilon^3)$. It is demonstrated how this approach can be used to simulate the HHL performance with a large number of variables, and the runtime of this approach is compared with existing quantum and classical methods for both solving linear systems of equations and simulating the HHL algorithm classically. The main contributions of this work are the following.

- The formulation of a qudit HHL algorithm.

- The formulation and implementation of a classical tensor network algorithm to classically simulate the best expected performance of the HHL algorithm.

- Comparison of this algorithm against well-known classical algorithms in different scenarios.

- A numerical analysis of the sensitivity of the HHL algorithm with respect to its hyperparameters.

These results show that this approach can achieve a promising performance in computational efficiency to simulate the HHL process without quantum noise. However, it does not improve the performance of the state-of-the-art best-known linear solver algorithms, and it is not intended to do so.

The paper is organized as follows. Initially, Sec. 2 presents a concise overview of classical algorithms designed to solve linear equations, as well as the HHL algorithm, omitting detailed quantum computing concepts, and an innovative qudit algorithm in qudits that enhances the original HHL is introduced. Following this, Sec. 3 describes the classical tensor network approach to simulate the optimal HHL performance and analyzes its complexity. Sec. 4 is devoted to comparing the novel algorithm with the conjugate gradient method, the traditional HHL, and the statevector simulation of the HHL. In the final section, in Sec. 5 the algorithm is applied to three simulation problems, contrasting it with an exact inversion algorithm, its performance is evaluated relative to an implementation of the HHL algorithm simulated via the Qiskit framework, and its sensitivity of the HHL performance respect to its hyperparameters. All code is available in the GitHub repository https://github.com/DOKOS-TAYOS/Tensor_Networks_HHL_algorithm.git, it can be openly tested online on the Streamlit website https://tensornetworks-hhl-algorithm.streamlit.app/, and there is a brief spanish explanation of the paper in the video Algoritmo HHL con Tensor Networks y qudits.

## 2 Background

There exist several algorithms to solve systems of linear equations, but only a few interesting ones will be introduced. All of them solve the system of linear equations

$$A\vec{x} = \vec{b}, \tag{1}$$

94 where $A$ is an invertible matrix $N \times N$, $\vec{x}$ is the vector to be obtained, and $\vec{b}$ is another vector,
95 both of dimension $N$.

96 The first is *Gaussian elimination*, which consists of reducing the augmented matrix $[A|b]$
97 to a matrix of the row echelon through row addition operations and transforming it into a di-
98 agonal matrix with the required solutions. Its computational complexity is $\mathcal{O}(N^3)$. It is widely
99 used in small cases, but when matrices become too large, other methods are applied because of
100 its prohibitive cubic cost [18]. Another extended algorithm is the *LU decomposition* [1], which
101 can be viewed as another way to perform Gaussian elimination. It decomposes the matrix $A$
102 into two matrices $L$ (lower triangular) and $U$ (upper triangular) that satisfy $A = LU$. It has
103 the same complexity as Gaussian elimination. Following the same line, there is the *Cholesky*
104 *decomposition* [19], which consists of decomposing a Hermitian positive definite matrix $A$ into
105 a lower triangular matrix $L$ with real positive diagonal entries that satisfy $A = LL^*$. Its com-
106 plexity is $\mathcal{O}(N^3)$, but it has half the cost of LU decomposition. This method is widely used for
107 Monte Carlo simulation, but not in general inversion due to the Hermitian positive definite
108 matrix restriction.

109 In the field of iterative methods, the most simple one is the *Jacobi algorithm* [20]. First,
110 the matrix $A$ is decomposed into three matrices $D$ diagonal, $L$ lower triangular, and $U$ upper
111 triangular such that $A = D + L + U$. For the $k$-th iteration, the transformation

$$\vec{x}^{(k+1)} = D^{-1}\left(\vec{b} - (L + U)\vec{x}^{(k)}\right) \tag{2}$$

112 is applied until it converges. Its complexity is $\mathcal{O}(N^2)$ for each step in the dense case and can
113 take advantage of the sparsity of $A$, but the number of necessary steps is not well-known. For
114 this reason, it is used in large problems.

115 An improvement of this method is the *Gauss-Seidel* algorithm for positive definite matrices,
116 or strictly diagonally dominant matrices, with no zero diagonal elements. In this case, the
117 iteration is made with

$$x_i^{(k+1)} = \frac{1}{a_{ii}}\left(b_i - \sum_{j=1}^{i-1} a_{ij}x_j^{(k+1)} - \sum_{j=i+1}^{n} a_{ij}x_j^{(k)}\right). \tag{3}$$

118 This method has the advantage in the storage, allowing one to rewrite the first iteration vector,
119 which enables to address larger problems, but it is much harder to implement in parallel.

120 Another iterative algorithm is the well-known *conjugate gradient method* [4] (CG) for large
121 sparse positive semidefinite matrices. Two non-zero vectors $\vec{v}$ and $\vec{w}$ are conjugate with respect
122 to $A$ if

$$\vec{v}^T A \vec{w} = 0. \tag{4}$$

123 If a set $P = \{\vec{p}_0, \vec{p}_1, \ldots, \vec{p}_{N-1}\}$ of mutually conjugate vectors with respect to $A$ is defined, then
124 $P$ forms a basis on $\mathbb{R}^N$ and

$$\vec{x} = \sum_{i=0}^{N-1} \alpha_i \vec{p}_i \implies A\vec{x} = \sum_{i=0}^{N-1} \alpha_i A\vec{p}_i. \tag{5}$$

125 Multiplying the problem by the left with $\vec{p}_j^T$

$$\vec{p}_j^T \cdot \vec{b} = \vec{p}_j^T A\vec{x} = \sum_{i=0}^{N-1} \alpha_i \vec{p}_j^T A\vec{p}_i = \alpha_j \vec{p}_j^T A\vec{p}_j, \tag{6}$$

126 then

$$\alpha_j = \frac{\vec{p}_j^T \cdot \vec{b}}{\vec{p}_j^T A\vec{p}_j}. \tag{7}$$

So, a set $P$ is needed, and then the $\alpha_j$ values are computed. However, if good $\vec{p}_i$ vectors are chosen, it may not be necessary to compute all of them to solve the problem. So, the problem will be solved approximately. The algorithm starts with an initial guess $\vec{x}_0$, and has a function to minimize

$$f(\vec{x}) = \frac{1}{2}\vec{x}^T A \vec{x} - \vec{x}^T \cdot \vec{b}. \tag{8}$$

Due to $\nabla f(\vec{x}) = A\vec{x} - \vec{b}$, it takes $\vec{p}_0 = \vec{b} - A\vec{x}_0$. $\vec{r}_j$ being the residual of the $j$-th step, and the negative gradient of $f$ in $\vec{x}_j$

$$\vec{r}_j = \vec{b} - A\vec{x}_j. \tag{9}$$

To obtain a vector $\vec{p}_j$ conjugate to the other ones, the transformation

$$\vec{p}_j = \vec{r}_j - \sum_{i<j} \frac{\vec{r}_j^T A \vec{p}_i}{\vec{p}_i^T A \vec{p}_i} \vec{p}_i, \tag{10}$$

is applied, and $\vec{x}_{j+1} = \vec{x}_j + \alpha_j \vec{p}_j$ is updated with

$$\alpha_j = \frac{\vec{p}_j^T \cdot (\vec{b} - A\vec{x}_j)}{\vec{p}_j^T A \vec{p}_j} = \frac{\vec{p}_j^T \cdot \vec{r}_j}{\vec{p}_j^T A \vec{p}_j} \tag{11}$$

until the residual is small enough.

Similar methods such as the *generalized minimal residual method* [21] and the *biconjugate gradient stabilized method* [22] are based on similar ideas as improvements for more general problems. All of these algorithms require matrix-vector multiplications, so their complexity is at least linear in the dimension of the problem. However, these faster state-of-the-art algorithms are only for specific problems, so they are presented only for extreme performance comparison against the following HHL algorithm. This is the reason because the common use libraries as NumPy or PyTorch make use of LU factorization for inversion problems to be able to solve general problems.

In this context, the *quantum linear solver algorithm* HHL [6] provides a computational advantage. Now, the standard HHL algorithm in qubits is introduced in order to better understand the algorithm that will be formalized.

For this algorithm, $n = \lceil \log_2 N \rceil$ qubits are needed to encode the vector $\vec{b}$, $n_c$ clock qubits to encode the possible eigenvalues of $A$ and one auxiliary qubit for the inversion with respect to these eigenvalues. The whole circuit can be summarized in Fig. 1a.

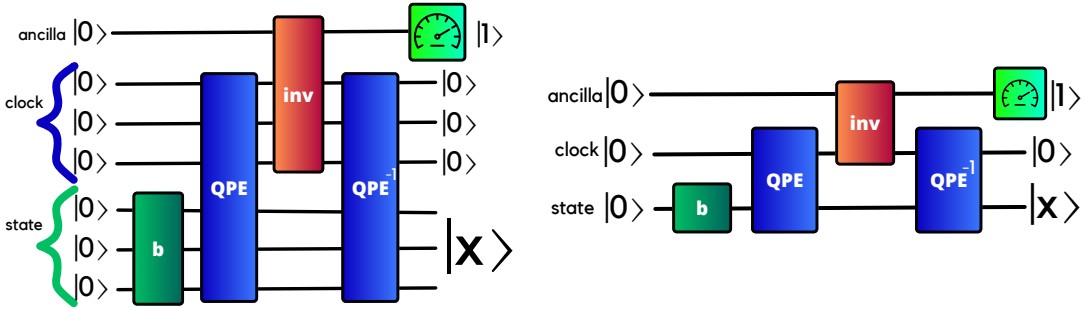

(a) Quantum HHL in qubits with $n = n_c = 3$.                   (b) Quantum HHL in qudits.

Figure 1: Comparison of HHL algorithm implementations.

The state $\vec{b}$ is encoded in the amplitudes of the $n$ state qubits

$$|b\rangle = \sum_{i=0}^{2^n-1} b_i |i\rangle = \sum_{i=0}^{2^n-1} \beta_i |u_i\rangle, \tag{12}$$

where $b_i$ are the normalized components of the vector $\vec{b}$, $|i\rangle$ are the computational bases states and $|u_i\rangle$ the eigenvector associated with the eigenvalue $\lambda_i$ of $A$. An operator $b$ is applied to initialize it. It is important to note that the difference between $N$ and $2^n$ will be completed with zeros in the vector and a matrix proportional to the identity in $A$, wasting resources. Moreover, a method is needed to generate this state $|b\rangle$ or the $b$ operator [23, 24].

The second step is to calculate the evolution operator

$$U = e^{iA'\tau}, \tag{13}$$

$\tau$ being a hyperparameter to rescale the eigenvalues for the next step and the matrix $A' = A$ if $A$ is Hermitian, and

$$A' = \begin{pmatrix} 0 & A \\ A^\dagger & 0 \end{pmatrix} \tag{14}$$

if it is not. In this case, the problem is

$$A' \begin{pmatrix} 0 \\ \vec{x} \end{pmatrix} = \begin{pmatrix} \vec{b} \\ 0 \end{pmatrix}. \tag{15}$$

It is assumed that $U$ can be calculated and implemented efficiently [25].

With this operator $U$ presented, a *Quantum Phase Estimation* (QPE) [26] is performed to encode the eigenvalues of $A'$ in the clock qubits. Since there are $n_c$ qubits, $\mu = 2^{n_c}$ possible values of eigenvalues can be encoded. For this reason, $n_c$ (and therefore $\mu$) is a hyperparameter for resolution, since it has to be chosen based on the properties of the problem, usually the conditional number of the matrix. Typically, $n_c = O(\log_2(\kappa/\epsilon))$, then $\mu = O(\kappa/\epsilon)$ [6].

Now, an inversion operator is applied, which rotates the probability of the auxiliary qubit so that it is divided by the value of the eigenvalue encoded by the QPE.

The next step is to make a post-selection, keeping only the state if the auxiliary qubit outputs a $|1\rangle$, followed by an inverse QPE to clean the eigenvalue qubits.

In the end, the result is the $\vec{x}$ state normalized in the amplitudes,

$$|x\rangle = \frac{1}{\mathcal{N}} \sum_{i=0}^{2^n-1} \frac{\beta_i}{\lambda_i} |u_i\rangle = \frac{1}{\mathcal{N}} \sum_{i=0}^{2^n-1} x_i |i\rangle, \tag{16}$$

with a normalization constant $\mathcal{N}$ and omitting the ancilla and clock qubits.

To obtain the full state vector, one needs to obtain the probability distribution, which requires measuring so many times that the computational advantage is lost. For this reason, the HHL is intended to obtain the expectation value of some operator with respect to the solution, or to introduce the solution state into another quantum subroutine.

The main problems of the algorithm are:

1. Large amount and waste of resources due to the difference between the size $N$ of the problem and the $n$ qubits to encode it.

2. Circuit depth and errors introduced by the state preparation and the QPE, in addition to the need for probabilistic post-selection.

3. $\vec{x}$ is not explicitly obtained and if it is extracted, it is obtained with a sign ambiguity for each of its elements.

4. The preparation of state $\vec{b}$ may not be trivial, just as performing the inversion operator or performing the $U$ operator.

To try to overcome the first two problems of the HHL, a qudit version of the algorithm is formalized. The main assumption for this algorithm is the existence of quantum computers that implement the basic qudit gates as described in the paper [8]. This assumption is basic, since our work is only theoretical, without going into all the technical difficulties of the hardware. This section is simply a mathematical motivation for the last tensor network algorithm.

The first step is to encode the state $\vec{b}$ in a single qudit. In case the qudit does not have enough states available, the vector is encoded in a number of qudits that allows to encode it in a way analogous to the case of qubits. In the following, it is assumed that only one qudit with $N$ basis states is needed in order to clearly explain the algorithm.

It is important to understand that the goal of this encoding is to reduce the physical amount of quantum resources. It is needed to use a Hilbert space exponentially larger than each of the Hilbert spaces of dimension 2 that would be used in the original case. However, it is assumed that there are physical systems that have this number of states available, since this is the basic premise of the work. An example would be to have a particle with a quantum number with possible values $-3/2, -1/2, 1/2, 3/2$ instead of another particle with possible values $-1/2, 1/2$.

Moreover, this change in the space used for the encoding will not imply a loss of the computational advantage of the original algorithm in the case of being able to realize the gates expressed in the paper [8], since the unitary gate $U$ is applied over the entire set of state qubits, as if it were a matrix over an ordinary vector.

Now, a way to simulate the $U = e^{iA'\tau}$ operator is needed, which depends on the particular case to be solved. With this, the following circuit in Fig. 1b is performed. With a single qudit of dimension $\mu = 2^{n_c}$ the QPE can be performed as in [8] and encode the $\mu$ possible values of the eigenvalue in its basis states. However, more qudits can be used. If 2 qudits are used to encode the $\mu$ values, each one will need to have dimension $\sqrt{\mu}$. This change in resources from clock qubits to a clock qudit will not imply a loss of quantum advantage either, since the algorithm will have the same number of possible eigenvalues.

The inverter is exactly the same as in the case of qubits, but instead of having a control-noncontrol series, there is a control $i$ that applies the rotation gate to the ancilla if there is a value $i$ in the qudit.

The post-selection is performed and if it outputs $|1\rangle$, the inverse QPE is performed to clear the qudit of the eigenvalues.

With this, the number of SWAP gates needed is reduced and the QPE is performed with a low number of gates. In addition, less resources are wasted, as the dimensionality of the quantum system can be better adjusted with respect to the equation to be solved, without the need to have extra elements at 0 to complete the $2^n$ components, which requires an extra qubit. Moreover, in the best-case scenario only two qudits and one qubit are needed, and in the worst-case scenario the same resources as in the original HHL are needed.

The classical simulation of both algorithms can be performed with statevector simulators. However, this simulation can be resource intensive with respect to the size of the problem. In the case of qubits, $n + n_c + 1$ qubits are required, so the storage of its statevector has a spatial complexity of $\mathcal{O}(2^{(n+n_c)})$, which in terms of $N$ and $\kappa$ is $\mathcal{O}(N\kappa/\epsilon)$. The state preparation usually requires $\mathcal{O}(2^{2n}) = \mathcal{O}(N^2)$ operations, due to the application of $\mathcal{O}(2^n)$ dense operations to a dense statevector [23, 24], and then extended for the $n + n_c + 1$ qubits with cost $\mathcal{O}(2^{(n+n_c)}) = \mathcal{O}(N\kappa/\epsilon)$. However, it can also be initialized directly without other operations and simply extended, reducing the cost of $\mathcal{O}(N^2)$ to $\mathcal{O}(N)$. However, the extension will not be applied in this step, to make the QPE in a more efficient way.

The controlled part of QPE requires the application of $n_c$ controlled gates $U^j$. In conventional statevector simulators, which work with dense statevector operations, the $\hat{m}$-th operation would require $\mathcal{O}(2^{(2n+\hat{m})})$ operations, because the statevector is extended only to the

clock qubits that are involved in every step, so the total $n_c$ steps require

$$\mathcal{O}\left(\sum_{\hat{m}=1}^{n_c} 2^{(2n+\hat{m})}\right) = \mathcal{O}\left(2^{(2n+n_c)}\right) = \mathcal{O}\left(N^2\kappa/\epsilon\right)$$

operations [6]. If the $U$ gate is decomposed in $\mathcal{O}(\log(N)s\tau)$ depth [27], requiring

$$\mathcal{O}\left(2^{n+n_c}\log(N)s\tau\right) = \mathcal{O}\left(N\kappa/\epsilon\log(N)s\tau\right).$$

Since $\tau = O(\kappa/\epsilon)$ [6], the computational complexity is $\mathcal{O}\left(N\log(N)\kappa^2/\epsilon^2 s^2\right)$. However, the inverse QPE does not allow for iterative extension, so it always requires the computation with all the statevector. This means that its computational complexity is $\mathcal{O}\left(2^{(2n+n_c)}n_c\right) = \mathcal{O}\left(N^2\kappa/\epsilon\log(\kappa/\epsilon)\right)$ for the dense $U$ and $\mathcal{O}\left(2^{n+n_c}n_c\log(N)s\tau\right) = \mathcal{O}\left(N\log(N)\kappa^2/\epsilon^2\log(\kappa/\epsilon)s\right)$ for the decomposed.

Inverse QFT requires the application of $\mathcal{O}\left(n_c^2\right)$ operations, so the total computational complexity of this step is $\mathcal{O}\left(2^{(n+n_c)}n_c^2\right) = \mathcal{O}\left(N\kappa/\epsilon\log^2(\kappa/\epsilon)\right)$, negligible with respect to the previous ones.

The inversion circuit can be applied in two main ways. The first is the naive multicontrolled rotations circuit without decomposition, which applies $\mathcal{O}(2^{n_c})$ rotations to every possible value of the eigenvalues, controlled by $n_c$ qubits. In conventional statevector simulators, this requires $\mathcal{O}\left(2^{(n+2n_c)}\right)$ operations for each eigenvalue, resulting in $\mathcal{O}\left(2^{(n+3n_c)}\right) = \mathcal{O}\left(N\kappa^3/\epsilon^3\right)$. However, in optimized statevector simulators, the fact that these are controlled operations can be exploited to reduce the complexity per eigenvalue to $\mathcal{O}\left(2^{(n+n_c)}\right)$, resulting in $\mathcal{O}\left(2^{(n+2n_c)}\right) = \mathcal{O}\left(N\kappa^2/\epsilon^2\right)$. The second way consists in the decomposition of each one of these multi-controlled rotations into $\mathcal{O}(n_c)$ one- and two-qubit gates, resulting in $\mathcal{O}\left(2^{(n+2n_c)}n_c\right) = \mathcal{O}\left(N\kappa^2/\epsilon^2\log(\kappa/\epsilon)\right)$ operations.

All of this implies that the complete computational complexity for simulating the statevector output of the HHL with common statevector simulators has a computational complexity of

$$\mathcal{O}\left(N^2\kappa/\epsilon\log(\kappa/\epsilon) + N\kappa^2/\epsilon^2\log(\kappa/\epsilon)\right),$$

neglecting the decomposition and transpilation complexities. With optimized statevector simulators it can be improved to

$$\mathcal{O}\left(N^2\kappa/\epsilon\log(\kappa/\epsilon) + N\kappa^2/\epsilon^2\right).$$

However, all of this process implies the usage of complex decomposition techniques, which incorrectly implemented may increase the computational complexity of the simulation. In regular implementations, without these complex techniques, the computational complexity will remain

$$\mathcal{O}\left(N^2\kappa/\epsilon\log(\kappa/\epsilon) + N\kappa^3/\epsilon^3\right).$$

Still, more can be done to solve the other problems, so the quantum-inspired technique of tensor networks will tackle them, avoiding gate errors from quantum devices and extracting $\vec{x}$, and allowing for a more efficient simulation of the algorithm.

# 3   Tensor Networks Algorithm

To return the vector $\vec{x}$ directly, the qudit circuit is transformed into a tensor network. This algorithm is called the *tensor network HHL* (TN HHL). Since in tensor networks normalization is not necessary, the state $|b\rangle$ will not be normalized. As it is not a unit vector, the result state is not normalized either, so it is not necessary to rescale it. Moreover, the state can be prepared exactly in a single operation, defining the node $\vec{b}$ with dimension $N$.

257    The QPE is performed by contracting the uniform superposition clock state with the *Quan-*
258  *tum Fourier Transform* (QFT) [28] in the QPE, so it is replaced by a matrix $H[\mu]$ with dimension
259  $\mu \times \mu$ for the $\mu$ eigenvalues with elements

$$H[\mu]_{ab} = e^{2\pi i \frac{ab}{\mu}}, \tag{17}$$

260  without normalization.

261    The inverter is a non-unitary operator with dimension $\mu \times \mu$ for $\mu$ eigenvalues whose non-
262  zero elements are

$$\text{inv}[\mu]_{i,j} = \begin{cases} 1/i & \text{if } i = j \neq 0 \text{ and if } i \leq \frac{\mu}{2} \\ \frac{1}{i-\mu} & \text{if } i = j \text{ and if } i > \frac{\mu}{2} \end{cases} \tag{18}$$

263  in order to encode negative eigenvalues due to the cyclic property of the imaginary exponen-
264  tial. If more positive or more negative eigenvalues are necessary, the proportion of $i$ values
265  that are translated as positive or negative eigenvalues must be changed.

266    The phase kickback operators can also be obtained exactly from $U$. This tensor $P$ with
267  dimension $N \times \mu \times N$ is

$$P[\mu]_{i,j,k} = \left(U^j\right)_{i,k}; \left(P[\mu]^{-1}\right)_{i,j,k} = \left(\left(U^{-1}\right)^j\right)_{k,i}. \tag{19}$$

268  These tensors are contracted through their $j$ index with the $H[\mu]$ and $H[\mu]^{-1}$ tensors to per-
269  form QPE.

270    With these tensors, the result can be obtained by contraction of the tensor network in Fig.
271  2 a

$$\sum_{a,b,c,d,e,f} b_a P_{abc} H_{bd}^{-1} \text{inv}_{de} H_{ef} P_{cfi}^{-1} = \frac{x_i \mu^2}{\tau}, \tag{20}$$

272  with $b, c, d, e, f \in [0, \mu-1]$ and $a, c, i \in [0, N-1]$, omitting $[\mu]$, since it is the same for all the
tensors.

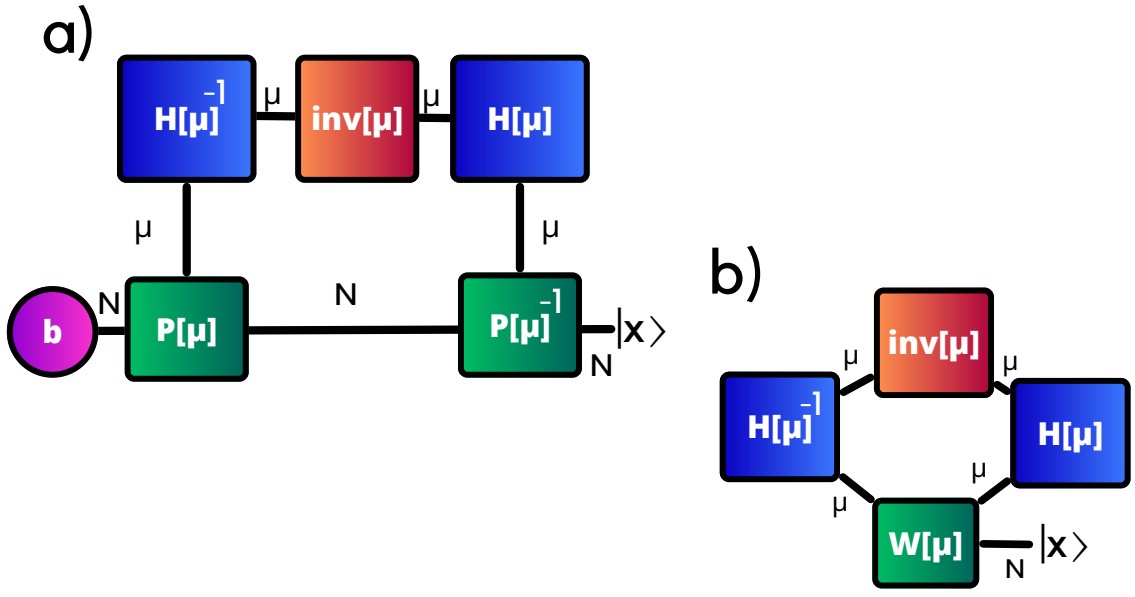

Figure 2: Tensor network equivalent to HHL. a) Original way. b) Efficient way.

273

274    It is assumed that contracting a tensor of $N$ indexes of dimension $n_i$ with another of $M$ in-
275  dexes of dimension $m_j$ through its first index has the usual computational cost of $\mathcal{O}\left(n_0 \prod_{i,j=1,1}^{N-1,M-1} n_i m_j\right)$.

The computational complexity of contracting the tensors $H[\mu]^{-1}$ and $\mathrm{inv}[\mu]$ is $\mathcal{O}\left(\mu^2\right)$, due to the sparsity of the tensor $\mathrm{inv}[\mu]$. Contracting the resulting tensor with $H[\mu]$ has a complexity of $\mathcal{O}\left(\mu^3\right)$. The contraction of $\vec{b}$ with $P[\mu]$ has a complexity of $\mathcal{O}\left(N^2\mu\right)$. The contraction of the tensors resulting from the contraction of $H[\mu]^{-1}$, $H[\mu]$ and $\mathrm{inv}[\mu]$ and the contraction of $\vec{b}$ and $P[\mu]$ has a computational complexity of $\mathcal{O}\left(N\mu^2\right)$. And the final contraction with $P[\mu]^{-1}$ has a complexity of $\mathcal{O}\left(N^2\mu\right)$.

Therefore, the contraction of the tensor network has a computational complexity of $\mathcal{O}\left(N^2\mu + N\mu^2 + \mu^3\right)$.

The construction of the tensors has a complexity:

- $H[\mu]^{-1}$ and $H[\mu]$: $\mathcal{O}\left(\mu^2\right)$.

- $\mathrm{inv}[\mu]$: $\mathcal{O}\left(\mu\right)$.

- $\vec{b}$: $\mathcal{O}(N)$.

- $P[\mu]$ and $P[\mu]^{-1}$: $\mathcal{O}\left(\mu N^3\right)$, since $N \times N$ matrices have to be multiplied up to $\mu$ times.

Therefore, the computational complexity of the algorithm is $\mathcal{O}\left(N^3\mu + N\mu^2 + \mu^3\right)$.

The complexity of calculating $U$ depends on the sparsity of the $A'$ matrix you use and the method. In the case of having $s$ non-zero elements per column, the complexity of this multiplication is $\mathcal{O}(sN^2\log(\omega))$, with $\omega$ being a number related to the convergence of the method used. However, this is highly dependent on the matrices used. For example, for a tridiagonal matrix, it can be done in $\mathcal{O}(N^2)$. The complexity of this term is additive to the complexities of the rest of the algorithm (it is performed only once), and it is not the largest one, so it will be omitted. The calculation of $U^{-1}$ requires the same complexity.

This increase in complexity due to the construction of the tensors $P[\mu]$ and $P[\mu]^{-1}$ can be avoided by defining a tensor $W[\mu]$ that directly computes the contraction of both tensors with the vector $\vec{b}$. Its elements are

$$W[\mu]_{ijk} = \left(\vec{b}U^{i-j}\right)_k. \tag{21}$$

This tensor has dimension $\mu \times \mu \times N$, and its construction requires multiplying up to $m$ times a vector with a matrix, which can be $U$ or $U^{-1}$. To construct each element, the algorithm can take advantage of another previously calculated element. Instead of having to calculate $U^{i-j}$, $U$ is multiplied by the previously calculated vector $\vec{b}U^{i-j-1}$ or $U^{-1}$ is multiplied by $\vec{b}U^{i-j+1}$. Therefore, the complexity of creating the tensor is $\mathcal{O}\left(N^2\mu\right)$.

So, the tensor network in Fig. 2 b has to be contracted, representing the equation

$$\sum_{a,b,c,d} W_{abi}H_{ac}^{-1}\mathrm{inv}_{cd}H_{db} = \frac{x_i\mu^2}{\tau}. \tag{22}$$

The complexity of contracting tensors $H[\mu]^{-1}$, $H[\mu]$, and $\mathrm{inv}[\mu]$ is the same as before. Contracting this resulting tensor with $W[\mu]$ has a complexity of $\mathcal{O}\left(N\mu^2\right)$. Therefore, the total computational complexity is $\mathcal{O}\left(N^2\mu + N\mu^2 + \mu^3\right)$. However, if the contraction of $H[\mu]$, $H[\mu]^{-1}$ and $inv[\mu]$ is pre-calculated to be used every time, the $\mathcal{O}(\mu^3)$ term could be avoided.

Moreover, it can be slightly improved in the case of $\mu = 2_c^n$ contracting the $W[\mu]$ tensor with the QFT circuits instead of dense tensors. This contraction consists of $n_c^2$ contractions of a tensor with $\mathcal{O}\left(2^{(n+2n_c)}\right)$ elements with a tensor of one or two qubits, with a resulting complexity of $\mathcal{O}\left(N\mu^2\log^2(\mu)\right)$. Then, it is contracted with the $inv[\mu]$ tensor with complexity $\mathcal{O}\left(2^{(n+2n_c)}\right)$ because it is diagonal, and finally with the other QFT circuit. This makes the algorithm require only $\mathcal{O}\left(N^2\mu + N\mu^2\log^2(\mu)\right)$ operations, avoiding the cubic scaling. This is the TN HHL W with QFT. Another possibility is to first multiply $inv[\mu]$ with the two QFT circuits, each

with complexity $\mathcal{O}\left(2^{2n_c}n_c^2\right) = \mathcal{O}\left(\mu^2 \log^2(\mu)\right)$, and a final contraction with $W[\mu]$ with complexity $\mathcal{O}\left(2^{(n+2n_c)}\right) = \mathcal{O}\left(N\mu^2\right)$. Then, the total complexity is $\mathcal{O}\left(N^2\mu + N\mu^2 + \mu^2 \log^2(\mu)\right)$, again avoiding the cubic term. This is the TN HHL inv with QFT.

The spatial complexity is $\mathcal{O}(N\mu^2 + N^2)$, being the first term associated with the tensor $W[\mu]$ and the second term associated with the matrix $U$.

The inverse of $A'$ can also be computed simply by erasing the $b$ node from Fig. 2 a and performing the contraction, increasing by $N$ all complexities.

## 4 Comparison of advantages and disadvantages

The computational complexities of these algorithms in tensor networks are compared in Table 1 against the relevant classical solvers CG and LU, the quantum HHL and the state-vector simulations.

| Algorithm | Solution $\vec{x} = A^{-1}\vec{b}$ | Expectation value $\vec{x}^T M \vec{x}$ |
|---|---|---|
| CG | $\mathcal{O}(Ns\kappa \log(1/\epsilon))$ | $\mathcal{O}(Ns\kappa \log(1/\epsilon) + Ns'))$ |
| LU | $\mathcal{O}(N^3)$ | $\mathcal{O}(N^3)$ |
| HHL | - | $\mathcal{O}(\log(N)s^2\kappa^2/\epsilon^3)$ |
| Naive Statevector | $\mathcal{O}\left(N^2\left(\frac{\kappa}{\epsilon}\right)\log\left(\frac{\kappa}{\epsilon}\right) + N\left(\frac{\kappa}{\epsilon}\right)^3\right)$ | $\mathcal{O}\left(N^2\left(\frac{\kappa}{\epsilon}\right)\log\left(\frac{\kappa}{\epsilon}\right) + N\left(\frac{\kappa}{\epsilon}\right)^3\right)$ |
| Common Statevector | $\mathcal{O}\left(\left(N^2\left(\frac{\kappa}{\epsilon}\right) + N\left(\frac{\kappa}{\epsilon}\right)^2\right)\log\left(\frac{\kappa}{\epsilon}\right)\right)$ | $\mathcal{O}\left(\left(N^2\left(\frac{\kappa}{\epsilon}\right) + N\left(\frac{\kappa}{\epsilon}\right)^2\right)\log\left(\frac{\kappa}{\epsilon}\right)\right)$ |
| Optimized Statevector | $\mathcal{O}\left(N^2\left(\frac{\kappa}{\epsilon}\right)\log\left(\frac{\kappa}{\epsilon}\right) + N\left(\frac{\kappa}{\epsilon}\right)^2\right)$ | $\mathcal{O}\left(N^2\left(\frac{\kappa}{\epsilon}\right)\log\left(\frac{\kappa}{\epsilon}\right) + N\left(\frac{\kappa}{\epsilon}\right)^2\right)$ |
| **TN HHL** | $\mathcal{O}\left(N^2\left(\frac{\kappa}{\epsilon}\right) + N\left(\frac{\kappa}{\epsilon}\right)^2 + \left(\frac{\kappa}{\epsilon}\right)^3\right)$ | $\mathcal{O}\left(N^2\left(\frac{\kappa}{\epsilon}\right) + N\left(\frac{\kappa}{\epsilon}\right)^2 + \left(\frac{\kappa}{\epsilon}\right)^3\right)$ |
| **TN HHL W with QFT** | $\mathcal{O}\left(N^2\left(\frac{\kappa}{\epsilon}\right) + N\left(\frac{\kappa}{\epsilon}\right)^2\log^2\left(\frac{\kappa}{\epsilon}\right)\right)$ | $\mathcal{O}\left(N^2\left(\frac{\kappa}{\epsilon}\right) + N\left(\frac{\kappa}{\epsilon}\right)^2\log^2\left(\frac{\kappa}{\epsilon}\right)\right)$ |
| **TN HHL inv with QFT** | $\mathcal{O}\left(N^2\left(\frac{\kappa}{\epsilon}\right) + N\left(\frac{\kappa}{\epsilon}\right)^2 + \left(\frac{\kappa}{\epsilon}\right)^2\log^2\left(\frac{\kappa}{\epsilon}\right)\right)$ | $\mathcal{O}\left(N^2\left(\frac{\kappa}{\epsilon}\right) + N\left(\frac{\kappa}{\epsilon}\right)^2 + \left(\frac{\kappa}{\epsilon}\right)^2\log^2\left(\frac{\kappa}{\epsilon}\right)\right)$ |

Table 1: Computational times to obtain the solution of $A\vec{x} = \vec{b}$ and compute an expectation value $\langle x|M|x\rangle$. $s$ and $s'$ are the maximum number of non-zero elements per row of the $N \times N$ matrices $A$ and $M$.

### 4.1 Quantum, statevector and TN algorithms vs classical algorithms

The quantum HHL algorithm is faster in terms of $N$ than the other algorithms for the computation of the expectation value, but it is not an algorithm to extract the explicit solution. The HHL statevector simulation is significantly slower than the CG, but it is faster than the LU in situations where $\kappa/\epsilon$ is small enough. With more precision, if $\left(\frac{\kappa}{\epsilon}\right)\log\left(\frac{\kappa}{\epsilon}\right) < N$. The TN HHL algorithms are also slower than the CG, but faster than the LU if $\left(\frac{\kappa}{\epsilon}\right) < N$. These are reasonable situations because it makes more sense to use the HHL algorithm in situations where $\kappa/\epsilon \approx \mathcal{O}(\text{polylog}(N))$ [6]. However, the multiplicative constant in complexity keeps these algorithms slower than the LU solver.

### 4.2 Quantum HHL vs Statevector HHL vs TN HHL

The quantum HHL is much faster than its classical simulations for the expectation value computation. The faster statevector method is the Optimized, and the faster TN method is the TN HHL inv with QFT. Both algorithms have similar complexities, but the TN is faster if $\left(\frac{\kappa}{\epsilon}\right)\log\left(\frac{\kappa}{\epsilon}\right) < N^2$, which is a reasonable assumption. If the comparison is between the most

342 naive simulators, the TN HHL is always faster than the naive statevector and is faster than the
343 common statevector if $\left(\frac{\kappa}{\epsilon}\right)/\sqrt{\log\left(\frac{\kappa}{\epsilon}\right)} < N$, reasonable again.

## 5 Experiments

345 The effectiveness of the TN HHL algorithm is tested by solving three common simple problems
346 with the expected structure of the problems that are the target of the quantum HHL algorithm.
347 The forced harmonic oscillator, the forced damped oscillator, and the 2D static heat equation
348 with sources are solved and compared against the solution provided by the PyTorch solver,
349 which makes use of the LU decomposition algorithm. Then, the performance of the TN HHL
350 is compared against a Qiskit implementation of the naive statevector HHL for random sparse
351 matrices. Finally, the sensitivity of the quality of the HHL solution is studied with respect to
352 their hyperparameter values, making use of the TN HHL.

353    It is important to note that the comparison is about the resolution with respect to the
354 system of linear equations, not with respect to the original differential equation itself. This is
355 because this method solves the discretization of the problem, so errors due to the discretization
356 itself should not be taken into account. In each experiment, the hyperparameters are tuned
357 by hand.

358    All experiments are performed in CPU, with an Intel(R) Core(TM) i7-14700HX 2.10 GHz
359 and 16 GB RAM.

### 5.1 Forced harmonic oscillator

361 This is the most simple problem with a complex structure. The differential equation to be
362 solved is

$$\frac{d^2x}{dt^2} + \frac{k}{m}x = F(t) \tag{23}$$
$$x_0 = x(t=0); \qquad x_T = x(t=T)$$

363 where $F(t)$ is the external force dependent on time $t$. For experiments, a harmonic force
364 $F(t) = 9\sin(0.4t)$ is chosen to be strong enough to modify the evolution of the system and
365 with a frequency that does not synchronize with the system frequency.

366    A discretization with $n$ time steps is applied, i.e. $\Omega = -2 + \frac{k}{m}(\Delta t)^2$ and $F_j = (\Delta t)^2 9\sin(0.4j\Delta t)$.

$$\begin{pmatrix} \Omega & 1 & 0 & \cdots & 0 & 0 \\ 1 & \Omega & 1 & \cdots & 0 & 0 \\ 0 & 1 & \Omega & \cdots & 0 & 0 \\ \vdots & \vdots & \vdots & \ddots & \vdots & \vdots \\ 0 & 0 & 0 & \cdots & 1 & \Omega \end{pmatrix} \begin{pmatrix} x_1 \\ x_2 \\ x_3 \\ \vdots \\ x_n \end{pmatrix} = \begin{pmatrix} F_1 - x_0 \\ F_2 \\ F_3 \\ \vdots \\ F_n - x_T \end{pmatrix} \tag{24}$$

367 The result of inverting this system gives us the result in Fig. 3. As hyperparameters of the
368 algorithm of the method, $\mu = 2000$ and $\tau = 6000$ are chosen.

369    The root mean square error of our tensor network from the exact inversion was $1.8 \times 10^{-5}$
370 and took 407 ms to run, compared to 244 $\mu$s of the exact inversion method of PyTorch.

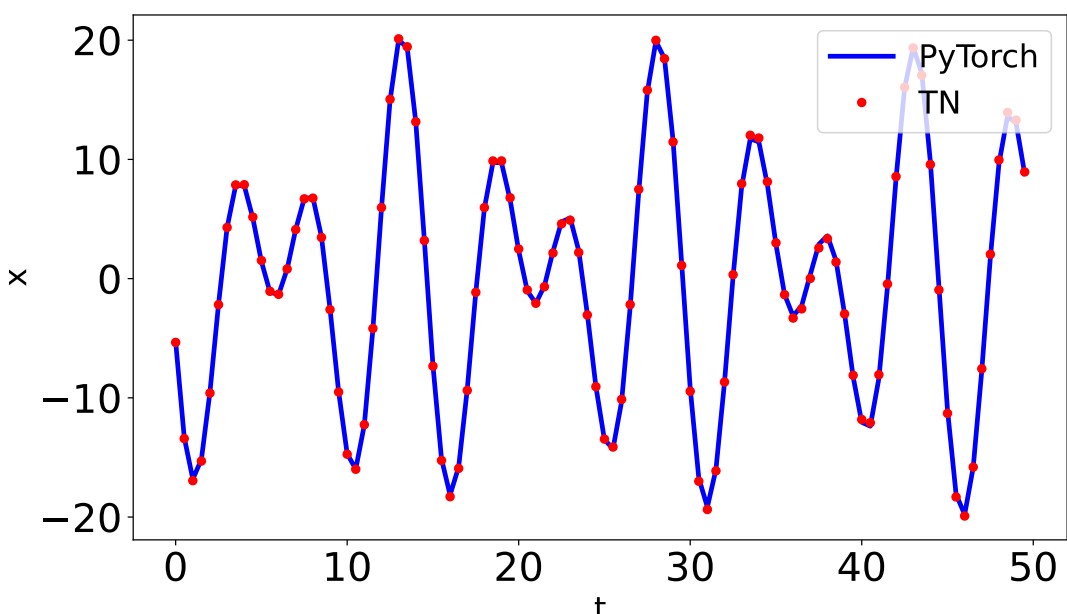

Figure 3: Solving the forced harmonic oscillator system with equation (24). In blue the inversion performed with PyTorch and in red the inversion performed with the tensor network. The parameters are $k = 5, m = 7, x_0 = 5, x_T = 3, \Delta t = 0.5, T = 50$.

## 5.2 Forced damped oscillator

This is a slightly more complex problem than the previous one. The differential equation to be solved is

$$\frac{d^2 x}{dt^2} + \gamma \frac{dx}{dt} + \frac{k}{m} x = F(t) \tag{25}$$
$$x_0 = x(t = 0); \qquad x_T = x(t = T)$$

where $F(t)$ is the external force dependent on time and $\gamma$ is the damp coefficient. As in Sec. 5.1, for the experiments, a harmonic force $F(t) = 9 \sin(0.4t)$ is chosen.

A discretization with $n$ time steps is applied

$$\begin{pmatrix} \beta_0 & \beta_+ & 0 & \cdots & 0 & 0 \\ \beta_- & \beta_0 & \beta_+ & \cdots & 0 & 0 \\ 0 & \beta_- & \beta_0 & \cdots & 0 & 0 \\ \vdots & \vdots & \vdots & \ddots & \vdots & \vdots \\ 0 & 0 & 0 & \cdots & \beta_- & \beta_0 \end{pmatrix} \begin{pmatrix} x_1 \\ x_2 \\ x_3 \\ \vdots \\ x_n \end{pmatrix} = \begin{pmatrix} F_1 - \beta_- x_0 \\ F_2 \\ F_3 \\ \vdots \\ F_n - \beta_+ x_T \end{pmatrix} \tag{26}$$

where $\beta_- = 1 - \gamma \frac{\Delta t}{2}$, $\beta_+ = 1 + \gamma \frac{\Delta t}{2}$ and $\beta_0 = -2 + \frac{k}{m}(\Delta t)^2$. This matrix is not hermitian, so (14) is applied and (15) is the equation to solve.

The result of inverting this matrix gives us the result in Fig. 4. As hyperparameters of the algorithm of the method, $\mu = 2000$ and $\tau = 1.1 \times 10^4$ are chosen.

The relative root mean square error of our tensor network from the exact inversion was $6.1 \times 10^{-3}$ and took 1.19 seconds to run, compared to 167 $\mu s$ of the PyTorch method.

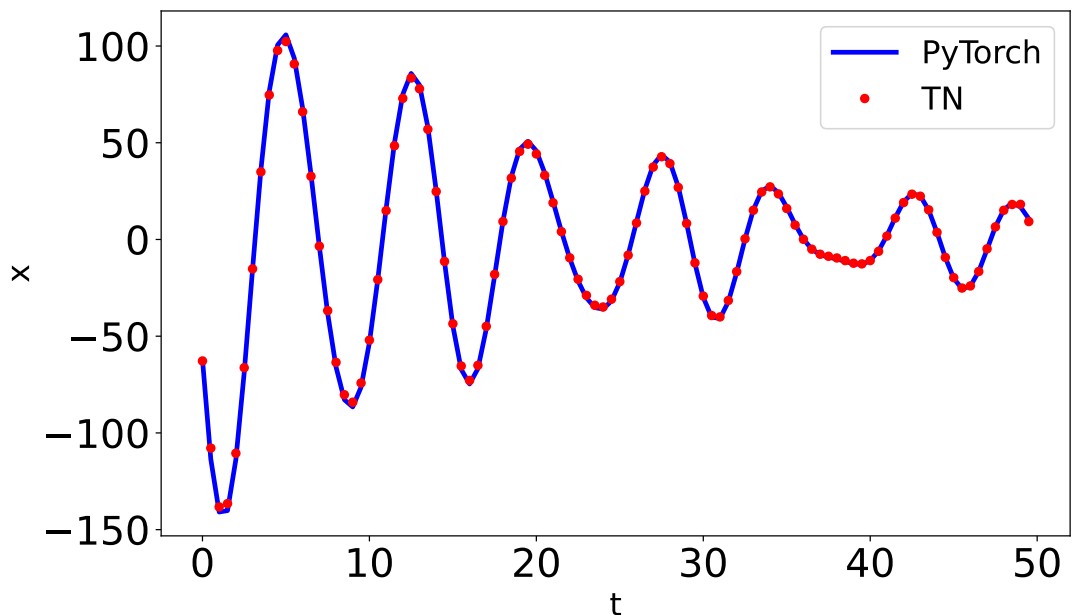

Figure 4:    Solving the forced damped oscillator system with equation (26).    In blue the inversion performed with PyTorch and in red the inversion performed with the tensor network.    The parameters are $k = 5, m = 7, \gamma = 0.1, x_0 = 5, x_T = 3, \Delta t = 0.5, T = 50$.

## 5.3   Static two dimensional heat equation with sources

This is a more complex problem than the two previous ones because of its bidimensional structure and interactions. The differential equation to be solved is

$$k\left(\frac{d^2u}{dx^2} + \frac{d^2u}{dy^2}\right) = -S(x,y) \tag{27}$$

$$u_{x1} = u(0,y); \quad u_{x2} = u(L_x, y)$$
$$u_{y1} = u(x,0); \quad u_{y2} = u(x, L_y)$$

where $S(x,y)$ is the external source dependent on position. For the experiments, a source $S(x,y) = 10\sin\left(2\pi\frac{xy}{L_xL_y}\right)$ is chosen to influence the system but not saturate its dynamics.

The chosen discretization is

$$u_{j+1,k} + u_{j-1,k} + u_{j,k+1} + u_{j,k-1} - 4u_{jk} = -(\Delta x)^2 S_{jk} \tag{28}$$

The 2-dimensional space is flattened into a line, create the matrix, and obtain the following result in Figs. 5a and 5b. As hyperparameters of the algorithm, $\mu = 2000$ and $\tau = 100$ are chosen.

The root mean square error of our tensor network from the exact inversion was $10^{-4}$ and took 4.64 seconds to run, compared to 1.98 ms of the PyTorch method.

## 5.4   Comparison against original HHL

Now, the results of the TN HHL are compared against an implementation of the original HHL algorithm in Qiskit, simulated in AerSimulator using the statevector backend. The experiments consists in the resolution of 20 random sparse matrices of dimension 16×16, due to the storage

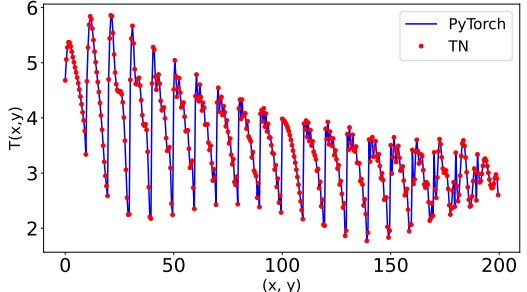
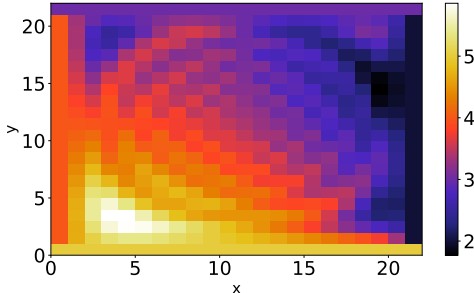

(a) Flattened view of the solution. In blue the inversion performed with PyTorch and in red the inversion performed with the tensor network.

(b) Bidimensional view of the solution.

Figure 5: Solving the static two dimensional heat equation with sources with equation (28). The parameters are $k = 3, u_{x1} = 5, u_{x2} = 3, u_{y1} = 4, u_{y2} = 2, \Delta x = 0.5, L_x = 10, L_y = 10$.

limitations of the statevector simulation. The matrices have a non-diagonal density of 25%, with all their elements generated in the interval $(-1, 1)$, the same for the vectors $\vec{b}$. After their generation, the matrices are transformed to symmetric matrices and normalized by their largest eigenvalue. For resolution, the values $n_c = 20$, $C = 1$, $\tau = 1$ and 100000 shots are chosen for original HHL and $\tau = 100$ are chosen for TN HHL, because they provide the best results for all methods. The correct result is also obtained by the PyTorch inversion function.

In the tests, the mean root mean square error obtained for the TN HHL is $7.5 \times 10^{-3}$, against $2.4 \times 10^{-2}$ for the original HHL. The mean root mean square error between the TN HHL and the original HHL is 1.047. In almost all of the experiments, the TN HHL obtained a better result than the original HHL. These results imply that the TN HHL avoids the errors of the original HHL to obtain a better solution. The TN HHL computes extremely faster than the naive statevector HHL simulation. The TN HHL needs 0.062 seconds per problem, compared to 17.766 seconds for the statevector. This is a 285 times speedup. The same experiment was intended with the matrix product state backend for tensor networks, but it is extremely slower than the statevector one, and gives worst results.

## 5.5 Sensitivity to hyperparameter tuning

Finally, this algorithm can be applied to study the sensitivity of the HHL algorithm respect to its hyperparameters $\tau$ and $n_c$. For this purpose, the same 20 random matrices are solved with the TN HHL for different values of $\tau$ and $\mu = 2^{n_c}$. Fig. 6 shows the results of the RMSE of the prepared vector $\vec{x}$ with respect to the PyTorch inversion, both normalized to keep that the result of the HHL must be a normalized vector.

Fig. 6a shows that the RMSE converges in small values of $\tau$ for all values of $n_c$, to the same value. It also shows that small $n_c$ implies an increase in the RMSE, limited by normalization, but for higher values it reduces until a minimum is reached for different values of $\tau$. It seems that every $n_c$ series follows the same convergence until they reach their saturation point, which has a higher $\tau$ and lower RMSE. Also, the saturation points seem to have a similar space from the previous one (in logarithmic scale), but for $n_c > 10$ they all have the same behavior. Fig. 6b shows that the increase in the number of clock qubits always decreases or maintains the RMSE until a saturation point. Again, the optimal amount of $n_c$ is not the same for all values of $\tau$, but increases with it, which has a lower RMSE. Both results show that the HHL performance is highly dependent on the hyperparameter values, but it may be predictable with regression

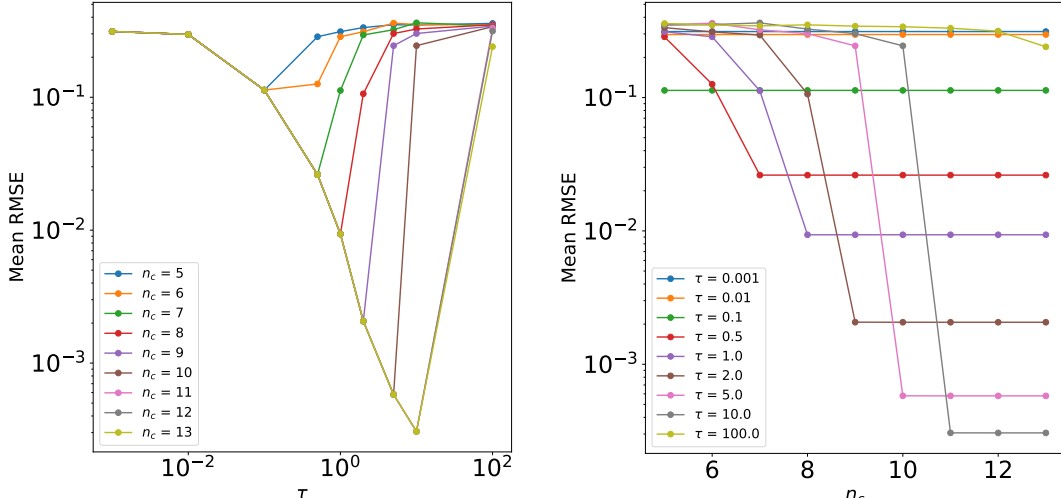

(a) Mean RMSE against the $\tau$ hyperparameter for each value of $n_c$.

(b) Mean RMSE against the $n_c$ hyperparameter for each value of $\tau$.

Figure 6: Comparison of Mean Root Mean Square Error (RMSE) against the hyperparameters of the HHL for random $16 \times 16$ matrices. Each point is the mean of the RMSE of 20 matrices.

techniques.

# 6 Conclusions

This work shows that this algorithm offers a way to efficiently simulate the HHL performance for solving systems of linear equations, invert matrices, and perform numerical simulations based on it. It has also been shown that its scaling is remarkably good with the size of the matrix to be inverted, while it can be realized on classical computers and accelerated with GPUs. This work also reports the existence of saturation points for the performance in the hyperparameters.

An advantage of this method is that it allows one to observe the best possible theoretical result due to a quantum HHL, as it simulates what should happen without gate errors, post-selection problems, or inaccuracies in statepreparation, providing a higher accuracy bound for the exact algorithm. This allows for its theoretical study prior to applying it in real noisy quantum devices or noisy simulators. However, the effective computational speed is remarkably low compared to methods already implemented in libraries such as PyTorch or Numpy for the resolution of the equations itself.

Future research in this area might aim to improve the overall efficiency of the method by leveraging the specific properties of the tensors used. For example, symmetry in the indices for eigenvalues in the $W[\mu]$ tensor may be applied to reduce the contraction complexity with the other tensors. This can also be particularized to tailoring the approach to tridiagonal matrices or adapting it for complex eigenvalues. Another possible line of research may be optimizing the parallelization of calculations or incorporating tensor network compression techniques to reduce runtime and storage requirements. Finally, an interesting line may be to apply this algorithm to a deeper study of the saturation points reported and the determination of a theoretical or empirical law that provides the optimal configuration of the HHL.

# Acknowledgements

**Funding information** The research leading to this paper has received funding from the Q4Real project (Quantum Computing for Real Industries), HAZITEK 2022, no. ZE-2022/00033.

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
