# Peer review of "Solving Systems of Linear Equations: HHL from a Tensor Networks Perspective"

_SciPost Physics_

## Round 1 · Referee Report · Anonymous (Referee 1) · 2025-8-19

Disclosure of Generative AI use

The referee discloses that the following generative AI tools have been used in the preparation of this report:

Asked GPT-5 specific questions about HHL and tensor network simulation of quantum algorithms. However, AI was not used to assist in the evaluation of the paper, nor in the writing of the report.

Strengths

1- Clearly written, the text flows smoothly

2- Honest account of what has been studied, how and what are some specific outcomes (but lacking vision, see weaknesses)

Weaknesses

1- No clear message: in which context should the proposed algorithm be used? What specifics alternatives should it be preferred over?

2- Some content appears irrelevant (eg details of classical algorithm against which theirs hasn't been benchmarked, redundant figures)

3- Comparison points are black boxes (Tensorflow, Qiskit Aer)

Report

In this paper, the authors consider a qudit version of the HHL algorithm, a key algorithm in quantum computing to solve linear systems of equations of the form Ax=b, which they turn into a tensor network simulation dubbed TN HHL.

It is stressed that this dequantised version of HHL does not aim to compete with efficient classical solvers such as the Conjugate Gradient method, as it is very slow.

Using TN HHL, the authors carry out matrix inversion on 3 example problems (forced H.O., forced damped H.O. and heat equation) in their discretised versions. They compare the resulting x (in terms of relative RMS) with that obtained with the exact inversion of A (Tensorflow, PyTorch). They find problem-dependent values ranging from about 10-5 to about 10-3, for a far superior simulation time (typically 10^3 times superior).

Then, they compare the results of TN HHL with HHL results obtained using Qiskit's Aer simulator. For this, they use random 16x16 matrices. They observe in average a slight increase in the quality of the solution using TN HHL rather than plain HHL.

I find the article to lack a clear direction and consistency. It seems that the authors simply do matrix inversion by contracting tensor networks in a way which is inspired from HHL. Errors arise as, just as HHL, the method is tuned by hyperparameters mu (number of eigenvalues) and tau (scaling parameter) but the effect of these on the accuracy over the solution is not studied - only best results are provided.

Here are more specific comments: 1- I feel like part II shouldn't be a part in its own as porting of HHL to qudits seems straightforward (which translates into eg Fig 2 being redundant of Fig.1) 2- Choices of physics examples to apply the method to are not explained (why forced, then forced+damped H.O. for instance? what is to be expected?) 3- The information content is confusing: the CG method to solve for x in Ax=b is thoroughly introduced in II, table I compares TN HHL with CG but the experiments are made against 'exact matrix inversion' instead (without any reference to what the actual algorithm running behind PyTorch and Tensorflow inverse methods is), Fig.7 is I believe another visualisation of data reported on Fig. 6, etc 4- Sometimes matrix inversion is made with PyTorch, another time with Tensorflow, without any mention of the reason behind this change in benchmarking tools 5- Some quantities are not introduced, such as c in eq. (17), hyperparameters C and t referenced in VI.D. 6- Many statements are qualitative and vague, see for instance terms flagged by * in the last sentence or the intro.: 'Our results show that our approach can achieve a promising performance in computational efficiency to simulate HHL process without quantum noise, providing a lower bound.' 7- In Sec VI.D., results are benchmarked against HHL run on qiskit on the aer simulator but the authors do not specify the backend. I assume this is the default, statevector simulator. Yet, there exists a tensor network backend in qiskit's aer. How does the specific TN HHL presented here compare to this one?

Requested changes

I believe the study is too preliminary for publication, and I'd encourage further investigation for a message to emerge.

If the goal is to advocate for the use of a TN HHL simulator to simulate the performances of HHL in a noise-free setting, then since HHL does not aim at providing x I believe the authors should aim for results in terms of operations over the solution.

As far as I know, a great strength of TN methods in the simulation of quantum algorithms lies in the possibility to mirror the effect of noise in the hardware by adapting the bond dimension (see eg https://journals.aps.org/prxquantum/abstract/10.1103/PRXQuantum.4.020304). This is advantageous since usual noisy simulation (eg via Lindblad) is very demanding in terms of resources.

The rationale behind using a TN method to simulate the noise-free behaviour of the QC with regards to the solution x, on the other hand, is not obvious to me. If there are specific types of errors that a noise-free quantum computation displays and that the TN simulation is able to capture with reduced simulation cost than statevector simulation, then it would be relevant to turn to TN to simulate HHL. This is however not argued nor shown in the paper.

On a side note, I spotted the following typos/English mistakes: - I believe the construction 'Ax=b, being A an invertible matrix' is grammatically incorrect and should be replaced eg by 'Ax=b, A being an invertible matrix' - 'eficient' in the caption of Fig. 3 (a) is missing an extra f - 'base' instead of 'basis' below eq (4) on p2 - 'Ssec' in VI. B

Recommendation

Ask for major revision

  • validity: low
  • significance: -
  • originality: -
  • clarity: low
  • formatting: excellent
  • grammar: good

Author:  Alejandro Mata Ali  on 2025-09-02  [id 5770]

(in reply to Report 1 on 2025-08-19)
Category:
remark
answer to question
objection

The main point of the algorithm is provide an efficient way to benchmark the HHL algorithm, because typical software tools and algorithms are slow in this task. For example, Statevector or MPS backends in Qiskit Aer are way slower, due to the large amount of operations with high dimensionality. Tensor Network backend directly fails a lot of times.

For the specific comments: 1. Yes, the qudit version does not need a subsection, may be included as part of the HHL explanation. 2. These examples are chosen to test the algorithm in toy problems that have matrix structures similars to the real problems that are intended to solve with the HHL, such as physics simulation. The most simple cases are these three. 3. PyTorch uses the LU factorization for the inversion of matrices. The CG is presented and compared to show the best known algorithm, but it is not general (as shown in the explanation), so it cannot be applied to these inversion problems. And yes, Fig. 7 is another visualization of Fig. 6. Maybe I must join them. 4. The Tensorflow is a typo from previous versions, all experiments are made with Pytorch, because we do not use GPU and it is not needed. 5. c in eq 17 is a typo, it must be mu. C and t are part of the differential equation definition, are the amplitude of the force and the time. 6. It is a lower bound for its error. This algorithm is always equal or better than the original one (due to the no-errors regime and avoids the post-selection). If this algorithm works bad, the HHL always must work even worse. 7. Yes, they are benchmarked against statevector, due to the first points of this reply.

For the request changes: - The point is to benchmark the provided state given by the HHL, because all the following of the algorithm depends on it. - The reason for using noise-free simulation is because, if the algorithm does not performs well without noise, it should be worse with it. Noise simulations with tensor networks are not that straightforward, and most of the time are not so interesting. In this case, a noisy simulation won't be the best idea because the main HHL does not perform well even without noise. Our simulation requires less resources than statevector for noise-free simulations, and this is the main focus of interest. - The typos and mistakes will be corrected.

---

## Round 1 · Referee Report · Anonymous (Referee 2) · 2025-9-1

Strengths

  1. Very well written.
  2. Codes are made available to all.
  3. Interesting ideas.

Weaknesses

The algorithm does not scale well relative to known approaches like conjugate gradient.

Report

The authors present a quantum-inspired classical algorithm to solve system of linear equations based on qudits and tensor networks. The authors extend the qubit HHL algorithm to qudit HHL by proposing that one can use 1 qudit for the state register, and 1 or more qudits for the clock register, with the aim of reducing quantum resources required for the circuit. This framework uses ideas from Ref. 8, and relies on qudit technology being available in the future. Then, they convert the entire qudit circuit into a tensor network (TN HHL) in order to extract x vector efficiently.

The work is interesting, it is very well-written and I enjoyed reading the work (especially the first two sections; the complexity parts are also nicely discussed, for example, lines 220-226), and it is very nice that the code is available to everyone for reproducibility, but I am not sure how the work advances the field of solving systems of linear equations efficiently (the authors of course do state this in the Introduction too). For example, Table I suggests that the complexity could even go as kappa times N^3 or more for A matrix inversion, which is more expensive that Jacobi or conjugate gradient (the authors are again very honest and tell this too explicitly in Section 5.1). Very very unfortunately, I cannot recommend publication in SciPost Physics Lecture Notes simply due to scaling.

Requested changes

A few optional very minor comments:
-Introduction paragraph 1: Is not kappa lambda_max/|lambda_min|?
-Introduction para 2: The cost of calculating expectation value using, say, the Hadamard test, would add further to the O(log(N)s^2kappa^2/epsilon), isn’t it? Is not the O(log(N)s^2kappa^2/epsilon) cost only for the HHL protocol?
-Line 81: What is n in n^3?
-Line 82: Maybe providing a reference would be helpful?
-Line 86: How do the number of iterations scale in system size for the Jacobi method?
-Line 88: definite positive —>? positive definite?
-TN HHL vs HHL section: Perhaps the authors can add that HHL scales much better in principle as it is a quantum algorithm with the log(N) benefit over classical approaches that have N or more?

Recommendation

Reject

  • validity: ok
  • significance: low
  • originality: good
  • clarity: top
  • formatting: excellent
  • grammar: excellent

Author:  Alejandro Mata Ali  on 2025-09-02  [id 5769]

(in reply to Report 2 on 2025-09-01)
Category:
remark
answer to question

The main point of the work is to provide an efficient way to simulate the best possible performance of the HHL, not to give an algorithm to improve the resolution of systems of linear equations, maybe I must to clarify it in a better way in future versions. This work is important because it provides a faster way to test and benchmark the HHL algorithm and, given the results, show that the HHL algorithm is not a good algorithm to use in an easy way, due to it low precision in the reality. If a algorithm that should be always better or equal in performance than the HHL without errors is still bad, the original algorithm must not be accurate in its real applications. Maybe I must to include more experiments for the dependence of the hyperparameters to show the bad performance of the original algorithm. Another important thing is that the algorithm is way faster than the direct implementation is Qiskit, with backends as the Aer Simulator one. This is the main motivation for the work. So the comparison must be 'This algorithm is better than the state-of-the-art for simulating the HHL', not 'This algorithm is better than the state-of-the-art for solving systems of linear equations'.

For the other comments: - Yes, I will correct the absolute value. - Yes, a factor 1/nu must be included. - It must be N, I will correct it. - I will include it. - To the best of our knowledge, is typically quadratic in N, but depends on the problem. - Yes, I will change it. - I will include it.

---

## Round 2 · Referee Report · Anonymous (Referee 1) · 2025-11-10

Report

Although I believe that this new submission constitutes an improvement, I still do not deem it suitable for publication. The new version seems to have suffered from an anchoring effect from the former one, so that the message is still not clear.

The remarks and suggestions contained in this report should not be taken as suggestions to simply iterate over this version of the draft, but rather as comments regarding how current results could have been better presented, and how to strengthen the study. Results could be presented in a whole new article, or in a heavily revised version of this one. By heavily revised, it is implied that things should be added but as least as importantly, that part of the content should be deleted or put in appendix.

General comments about the point the article is trying to make: - Since HHL is ruled out as an efficient way to obtain the explicit solution, but is rather thought as a way to access efficiently expectation values, I suggest the authors ‘state the obvious’ to justify that they still study how far from the exact solution their algorithm’s solution is. (namely, that if the distance between these two vanishes then the distance between the expectations values also vanishes) - The comparison to the exact solution is actually more a sanity check than anything. The main focus of the article should be the study of how HHL performs with respect to its hyperparameters, and how these impact the simulation time.

Regarding the new section (5.4), it would be better: - To plot $\tau_c = f(n_c)$, $\tau_c$ being the value of $\tau$ above which the mean RMSE starts going up as $\tau$ increases. This is a way to turn the rather convoluted statement ‘the saturation points seem to have a similar space from the previous one’ l. 423 into a quantitative statement. - Another thing which could be done would be the RMSE plotted as a colorplot against both $\tau$ and $n_c$. - To provide error bars (these are mean RMSE). Actually, were the 20 random matrices drawn constrained in terms of condition number? This number should at least be stored as it is crucial to the performance of the algorithm.

But I'd also suggest/note: - To carry out the study for relevant matrix inversion (such as problems studied in Sec 5.1/2/3), rather than just over random sparse matrices. - The way $\tau$ and $n_c$ should be picked is prescribed by the condition number $\kappa$ and the error $\epsilon$: $\tau=O(\kappa/\epsilon)$ (see l.232) and $n_c=O(\log_2 \kappa/\epsilon)$ (see l. 165). So the quantitative behaviour observed is to be expected and does not constitute a result per se. - Add data regarding the simulation time as hyperparameters are varied. This would render the TN implementation proposed here a tool of interest for resource estimation in the context of classical emulation of quantum algorithms.

Requested changes

I encourage the authors to pursue their study regarding the hyperparameters' dependence of HHL's performance, as this is where the scientific interest of their TN implementation lies. This study deserves to be carried out on relevant problem instances, and simulation times should be reported. Results should be added to a substantially trimmed version of the article.

Recommendation

Ask for major revision

---

## Round 2 · Author Response

This is a more precise and improved version. The main idea of the work is clearer and a deeper study of the statevector and the tensor network simulators is provided. Also, the main algorithm of the paper is improved, and new studies are performed. The code is also updated. Now, even a streamlit app is provided for full reproducibility. All the asked changes from the reviewers where performed, and the other concerns are solved. Thanks for the remarked points, they improved drastically the work.

---

## Round 2 · List of Changes

1. Several erratas and mistakes are corrected (the provided for reviewers and others).
  2. The abstract and introduction are changed to remark the importance of this work, as indicated by reviewers.
  3. A new link for a Streamlit app is provided for more reproducibility.
  4. New references for critical points of the work, as indicated by reviewers.
  5. The section of the qudits HHL is merged with the background section, as indicated by reviewers.
  6. PyTorch algorithm for inversion provided, as indicated by reviewers.
  7. Clearer definition of all variables and hyperparameters of the paper, as indicated by reviewers.
  8. From the line 223 to 250, a new analysis of the computational complexity of the statevector simulation of the HHL in different statevector configurations is provided. This is for clarifying the message, and it is novel in the literature.
  9. From line 310 to 319, two new versions of the algorithm are provided to improve its efficiency. This was a new discovery making the revision of the work.
  10. The table of the section 4 now includes the complexity of the statevector simulation in the three configurations, the LU, and the three provided TN HHL algorithms. Also the complexities are corrected.
  11. The section 4 is deeply changed. Now it compares the quantum, simulated and TN algorithms against the classical algorithms, and the quantum, simulated and TN one against the other. It is also provided the problems where each algorithm is better than other.
  12. Section 5 is clarified and the simulation experiments are slightly changed for more clarity. More information about them is provided for motivation, as indicated by reviewers.
  13. The comparison against statevector HHL now includes mean times of execution, and a reason for not using the matrix product state backend (extremelly slow), as indicated by reviewers. It can be verified in the new code.
  14. A new subsection of sensitivity analysis is included, as indicated by reviewers. Here new results about the HHL are provided, making use of the TN HHL.
  15. Conclusions have new future research lines.
  16. Minor style changes, transformation from active to pasive voice, and several changes in concrete explanations across the document for better understanding and motivation.

---

## Editorial Decision

in_refereeing